# Multi-Omics Analysis of the Anoikis Gene *CASP8* in Prostate Cancer and Biochemical Recurrence (BCR)

**DOI:** 10.3390/biomedicines13030661

**Published:** 2025-03-07

**Authors:** Shan Huang, Hang Yin

**Affiliations:** 1Department of Urology, Beijing Chao-Yang Hospital, Capital Medical University, Beijing 100020, China; 18661656191@163.com; 2Institute of Urology, Beijing Chao-Yang Hospital, Capital Medical University, Beijing 100034, China

**Keywords:** single-cell sequencing and spatial transcriptomics analysis, prostate cancer, biochemical recurrence

## Abstract

**Background:** Prostate cancer, as an androgen-dependent malignant tumor in older men, has attracted the attention of a wide range of clinicians. BCR remains a significant challenge following early prostate cancer treatment. **Methods:** The specific expression pattern of the Anoikis gene set in prostate cancer cells was first explored by single-cell and spatial transcriptomics analysis. Genes causally associated with prostate cancer were screened using Summary-data-based Mendelian Randomization (SMR). Subsequently, we explored the role and mechanism of *CASP8* in prostate cancer cells and defined a new cell type: the *CASP8* T cell. We constructed a prediction model that can better predict the BCR of prostate cancer, and explored the differences in various aspects of clinical subgroups, tumor microenvironments, immune checkpoints, drug sensitivities, and tumor-immune circulations between high- and low-risk groups. The results of SMR analysis indicated that *CASP8* could increase the risk of prostate cancer. Based on the differential genes of *CASP8*-positive and -negative T cells, we constructed a four-gene prognostic model with a 5-year AUC of 0.713. **Results:** The results revealed that high-risk prostate cancer BCR patients had various characteristics such as higher tumor purity, higher BCR rate, downregulated SIRPA immune checkpoints, and unique drug sensitivity. **Conclusions:** In summary, *CASP8* may be a potential biomarker for prostate cancer.

## 1. Introduction

Prostate cancer is a common androgen-dependent malignancy in men [1]. As the second most common cancer in men nowadays, it has been reported that one in six men in the UK are diagnosed with prostate cancer, with its early asymptomatic presentation and inert process being particularly prominent [2,3]. It is also worthy to discuss that there is significant interest in integrating image-based information from radiomics into the multi-omics framework, aiming to combine biomolecular-level information with imaging data. The clinical application of radiomics involves identifying the relationship between the features extracted from images and the clinical outcome of interest. In prostate cancer patients, common imaging modalities include MRI, transrectal ultrasound, conventional CT, cone-beam CT, and molecular imaging, often in the form of PET/CT with tracers such as radiolabeled Prostate-Specific Membrane Antigen (PSMA) and fluorine-labeled 18F-choline [4]. With the recent development of modern technology, the treatment of prostate cancer has developed a systematic and individualized process. Abiraterone with prednisolone combined with androgen deprivation therapy (ADT) should be considered a new standard treatment for patients with high-risk nonmetastatic prostate cancer. In a metastatic setting, enzalutamide and abiraterone should not be combined for those starting long-term ADT. Clinically important improvements in survival from the addition of abiraterone to ADT are maintained for longer than 7 years [5]. For early-stage limited prostate cancer, radical prostatectomy [6] or radiotherapy [7] is the first-line treatment choice with greater benefit. However, BCR following treatment remains a concern.

BCR is a topic of more concern after radical prostate cancer treatment, and its occurrence usually means recurrence or progression after prostate cancer treatment. There are two traditional definitions of BCR: (1) two consecutive PSA values above 0.2 ng/mL and rising after radical prostatectomy [8]; (2) elevated PSA >= PSA nadir of 2 ng/mL after radical radiotherapy [9]. We can see that PSA is still the best indicator for suggesting biochemical recurrence of prostate cancer. In recent years, many scholars have focused on the prediction of BCR in prostate cancer, hoping to know the BCR probability of patients earlier so as to achieve early prevention, early diagnosis, and early treatment and to have achieved certain results [10,11,12,13].

Up to now, programmed cell death has attracted widespread attention due to its multiple roles in the field of cancer. Anoikis, as an integrin-dependent mode of programmed cell death triggered by the loss of extracellular matrix and intercellular adhesion, has been significantly expressed in embryonic development, inflammatory response, tumor metastasis, etc. [14,15]. *CASP8*, an Anoikis gene, has an important role in a variety of cancers, including breast cancer [16,17], lung cancer [18,19], bladder cancer [20], and prostate cancer [21,22]. It has been reported that *CASP8*-652 6N ins/del polymorphism may affect the susceptibility to prostate cancer and reduce the risk of prostate cancer in Chinese men [21], while in other cancers it may be associated with cell death and tumor growth. Therefore, we explored the spatial-specific expression characteristics of Anoikis genes and *CASP8* in prostate cancer and the prognostic role in BCR of prostate cancer using multi-database and multi-omics data. In conclusion, this study innovatively constructed a prognostic model for the high- and low-risk groups of prostate cancer BCR by characterizing the signature genes of *CASP8*+ T cells and explored the clinical applicability of the model, which can help to guide clinical decision-making and provide personalized treatment for patients.

## 2. Methods

### 2.1. Data Download

The Anoikis gene set includes 338 genes in total, which can be obtained by downloading from the GeneCards database (https://www.genecards.org/ (accessed on 20 October 2024)). The single-cell analysis data were downloaded from the GEO database, and we selected the normal sample GSM5793824 and two tumor samples GSM5793828 and GSM5793829 from the GSE193337 data. The single-cell RNA-seq dataset was processed and analyzed using the R package “Seurat” (version 4.3.0). The spatial transcriptomic data were downloaded from the CROST database (https://ngdc.cncb.ac.cn/crost/ (accessed on 20 October 2024)) as expression matrix H5 files and Spatial folders, and the “Seurat” package was applied for subsequent spatial transcriptomic analysis.

### 2.2. Single-Cell and Spatial Transcriptomic Analysis of Anoikis Gene Set in Prostate Cancer

First, we merged the data from the two selected tumor samples and preprocessed the data: we excluded cells with fewer than 200 genes and more than 10,000 genes, as well as cells with mitochondrial gene proportions greater than 20% and ribosomal gene proportions greater than 20%, as these cells are considered to be of low quality. We normalized the data using the “LogNormalize” function and then used the “FindVariableFeatures” function to identify 3000 highly variable genes for subsequent PCA dimensionality reduction analysis. Second, we normalized all the genes and then performed PCA dimensionality reduction and clustering analysis on the data and displayed the cell cluster categories by applying the UMAP nonlinear dimensionality reduction algorithm. These preprocessed data were analyzed as the basis for a subsequent gene set and single gene analysis.

T cells were grouped into two categories based on gene set scores: T cells with high Anoikis expression and T cells with low Anoikis expression. Subsequently, cell communication and reverse chronology analysis were performed for 7 cell types of prostate cancer. Spatial transcriptome analysis was performed on the downloaded data using RCTD spatial deconvolution to obtain the expression of each cell type in the tissues and to analyze the interactions between the T cells with higher expression of Anoikis (AnoikishighT) and those with the highest correlation of cell communication. The “mistyR” package was utilized to explore whether the abundance of major cell types within a single transcriptome site could be predicted from their spatial context.

### 2.3. eQTL and Prostate Cancer Data Acquisition

We used SMR Portal (westlake.edu.cn) for SMR analysis. Expression quantitative trait loci (eQTL) data were obtained from eQTLGen (blood tissue, *n* = 31,684) and GTEx v8 eQTL (prostate tissue, *n* = 245; blood tissue, *n* = 755). We identified eQTL single-nucleotide polymorphisms (SNPs) that were significantly associated (*p* < 5 × 10^−8^) with the selected genes. eQTL were screened according to the SMR Portal default principle, and SNPs of the final 17,787 genes screened were subjected to subsequent SMR analysis, multi-SNP-based SMR test, and HEIDI test.

Prostate cancer data were obtained from the public database IEU OPEN GWAS (https://gwas.mrcieu.ac.uk/datasets/ (accessed on 20 October 2024)), and prostate cancer data with GWAS ID ieu-b-85 were selected and its VCF file was downloaded and organized. The dataset was obtained from the PRACTICAL website (http://practical.icr.ac.uk/ (accessed on 20 October 2024)) and included 140,254 individuals (79,148 cases and 61,106 controls), all in the European population. We used the collated data as outcome factors for subsequent analyses.

### 2.4. SMR Analysis of eQTLs for Prostate Cancer

SMR analysis is used to identify genetic loci that are causally associated with the outcome trait at the genetic level [23]. This analysis combines GWAS summary data with eQTL data to identify genes associated with expression levels of complex traits [24]. We used SMR Portal and identified eQTLs with causal associations with prostate cancer (p_SMR < 0.05 and p_HEIDI > 0.05). We took the intersection of eQTLs causally associated with prostate cancer and the set of Anoikis genes and screened for eQTLs associated with Anoikis and causally associated with prostate cancer. Finally, we chose the *CASP8* gene for subsequent analysis.

### 2.5. Single-Cell RNA Sequencing of CASP8 in Prostate Normal and Tumor Tissues

We combined the GSM5793824 normal sample and GSM5793828 tumor sample and then processed the data. First, the extracted data were quality controlled and standardized. Cells with unique feature counts of less than 200 and greater than 10,000, cells with mitochondria proportion greater than 20%, and cells with a ribosome proportion greater than 20% were excluded. The data were standardized using the “LogNormalize” function; then, we applied the “FindVariableFeatures” function to the data. The “FindVariableFeatures” function was used to obtain 2000 highly variable genes, which were used for subsequent PCA downscaling analysis. Second, after we normalized all the genes, combining with Jackstraw and Elbow plots, we determined the principal component (PC) to be 15 and further selected a resolution of 1.2 for PCA downscaling. Third, the UMAP nonlinear dimensionality reduction algorithm was applied to display cell clusters. Fourth, seven cell types (T cells; epithelial cells; endothelial cells; NK cells; monocytes; tissue stem cells; and smooth muscle cells) were assigned to the cell clusters according to the genetic markers by “SingleR”. Fifth, we analyzed and compared the high and low expression of *CASP8* in T cells of normal tissues and tumor tissues and scored the enrichment of the high- and low-expression groups of *CASP8* by calling the msigdbr package using the “irGSEA” package in “UCell” (version 2.6.2). Details of the preprocessed data are given in Appendix A.

### 2.6. Spatial Transcriptome Analysis of CASP8 in Prostate Cancer

We analyzed the preprocessed data from the tumor samples to study the expression level of *CASP8* in 6 cell types in the preprocessed data and grouped the T cells according to the high and low expression of *CASP8* in T cells, which were classified into *CASP8*-positive and *CASP8*-negative T cells. The cellular communication and spatial transcriptome analyses were performed on the seven finely delineated cell types; homotypic and heterotypic cell networks were constructed; and the co-localization of the cell types was described using the “mistyR” package.

### 2.7. Constructing and Evaluating Biochemical Recurrence (BCR) Prognostic Models for Prostate Cancer

The GSE116918 dataset was randomly divided into two groups: a training group and a validation group (Appendix A). We constructed a BCR prognostic model for prostate cancer based on the differential genes between *CASP8*+/− T cells and screened the core genes by one-way Cox regression, lasso regression, and multifactor Cox regression. Subsequently, the model was evaluated from multiple perspectives: (1) assessing the predictive ability of the model by time-dependent receiver operating characteristic (timeROC) and C-index quantification; (2) calibrating it by calibration curves; (3) constructing nomograms by combining clinical characteristics of the dataset; (4) performing decision curve analysis (DCA) on the factors in the nomograms.

### 2.8. Protein Expression Levels of CASP8 and Core Genes in Human Protein Atlas (HPA)

In order to verify the expression characteristics of genes in malignant tumors, we used immunohistochemistry data from Human Protein Atlas (HPA, http://www.proteinatlas.org (accessed on 20 October 2024)) to comparatively analyze the protein expression characteristics of *CASP8* and core genes in the prognostic model.

### 2.9. Gene Set Enrichment Analysis (GSEA)

GSEA is used to assess the trend of distribution of genes in a gene set of known function in a table of genes associated with a phenotype. We scored the gene expression matrix of the GSE116918 dataset for KEGG pathway enrichment by GSEA, where *p* < 0.05 was considered statistically significant. We performed GSEA on the high- and low-risk groups separately to obtain KEGG pathway scoring between the two groups.

### 2.10. Tumor Microenvironment and Immune Checkpoint Analysis

In addition to tumor cells, normal cells such as stromal cells and immune cells are also present in tumor tissues, and these cells also play an important role in cancer biology. Tumor purity refers to the proportion of cancer cells in a tumor sample. We applied the “ESTIMATE” package to calculate the stromal score and the immune score to infer tumor purity and to compare the difference between high-risk and low-risk BCR groups. Immune checkpoints refer to programmed death receptors and their ligands. Understanding the differences of immune checkpoints in the tumor microenvironment is helpful to start from the direction of checkpoints, restore the ability of T cells, and then kill tumor cells to improve the prognosis.

### 2.11. Drug Sensitivity Analysis and Immunotherapy

In this study, we applied the “oncoPredict” R package to analyze the Genomics of Drug Sensitivity in Cancer (GDSC) to obtain the differences in drug sensitivity between the high-risk group and the low-risk group so as to guide clinicians to select drugs and optimize the treatment plan.

### 2.12. Analysis of the Cancer–Immune Cycle

The anticancer immune response consists of seven events known as the cancer–immune cycle, which are (1) release of cancer cell antigens, (2) cancer antigen presentation, (3) priming and activation, (4) trafficking of immune cells to tumors, (5) infiltration of immune cells into tumors, (6) recognition of cancer cells by T cells, and (7) killing of cancer cells. Tracking Tumor Immunophenotype (TIP) is a meta-server that integrates “ssGSEA” and “CIBERSORT” methods to visualize the proportion of tumor-infiltrating immune cells during the cancer–immune cycle. In this study, the IOBR package was applied to visualize the differences between high- and low-risk groups in key steps of the cancer–immune cycle using the “ssGSEA” method.

### 2.13. Statistical Analyses

All the studies were analyzed in R software (version 4.3.0). A Pearson correlation test was performed when the data met the normality test. Otherwise, nonparametric tests (Wilcox rank sum test) were applied. *p* < 0.05 was considered statistically significant.

## 3. Results

### 3.1. Single-Cell and Spatial Transcriptome Analysis of the Anoikis Gene Set in Prostate Cancer

We scored T cells in the preprocessed data of the GSE193337 dataset for the Anoikis gene set and categorized T cells into those with high Anoikis expression (AnoikishighT) and those with low Anoikis expression (AnoikislowT) according to the median value of the score. The CellChat package was used to analyze the cellular communication between T cells and five other cell types by Secreted Signaling and Cell–Cell Contact, and we found that there was a strong cellular communication between AnoikishighT and monocytes (Figure 1A). Ligand–receptor interactions between T cells and monocytes were further analyzed, and we found a strong HLA-CD8 interaction between monocytes and AnoikishighT (Figure 1B). This provides a basis for developing novel immunotherapeutic strategies. For instance, modulating T-cell function by enhancing or inhibiting the HLA-CD8 interaction can improve antitumor immune responses. Subsequently, we performed reverse chronological analysis of the Anoikis gene set in T cells. First, we observed that the Anoikis gene set in T cells was highly expressed in cluster 0 by UMAP downscaling, and then we applied the “Vector” package to perform cell trajectory analysis to infer the differentiation trajectory of the cells during the developmental process. We calculated quantile polarization (QP) scores [25] to find cells at the beginning of development and further inferred the process of cell development. Figure 1C–E show that the region of high expression of the Anoikis gene set in T cells is located at the developmental end point, which may imply that the high-expression state of Anoikis genes builds up gradually with cell maturation, which suggests that Anoikis expression is reduced in the dedifferentiated tumor cells.

We analyzed the spatial transcriptome of the seven deconvolved cells by RCTD deconvolution to observe the spatial distribution characteristics in each cell type; details can be found in Appendix A. Figure 1F shows that the spatial distribution of AnoikilowT is similar to that of monocytes, and the spatial high-expression regions of AnoikishighT and monocytes are also somewhat linked. In conclusion, T cells and monocytes are closely linked in spatial distribution. Cells in space are influenced by their surroundings and themselves, resulting in a wide variety of cellular networks in their spatial locations. Homotypic cell networks are used to characterize the spatial distribution of a single cell type, while heterotypic cell networks are used to study the connectivity of different cell types in spatial locations. We constructed homotypic and heterotypic cell networks centered on AnoikishighT, respectively. Each spot in the network represents a cell type, and the spot degree of the six surrounding spots is 6 if all of them contain this cell type, 5 if five spots are of this cell type, and so on, which is divided into seven spot degrees. Homotypic and heterotypic cell networks were seen as regions of concentrated interactions between AnoikishighT homotypic cells and between AnoikishighT and monocytes. Enrichment score plots similarly confirmed the results for heterotypic cell networks (Figure 1G).

Co-localization analysis of various cell types was performed using the “mistyR” package. We defined three spatial scales, intra (0 spot radius), juxta_5 (5 spot radius), and para_15 (15 spot radius), and found that AnoikishighT had the highest R2 benefit (Figure 1H). The proportions of various cell types contributed differently at different spatial scales, but the proportions within the cell predominated (Figure 1I). The spatial dependence of AnoikishighT on monocytes was strongest within the spot and diminished with increasing distance, while AnoikislowT behaved in the opposite way to AnoikishighT (Figure 1J).

### 3.2. SMR Analysis of Causal Relationship Between eQTL and Prostate Cancer

In this study, we used SMR to analyze the potential causative genes for prostate cancer. We used a threshold of *p* < 5 × 10^−8^ to screen for eQTLs significantly associated with gene expression. To exclude false positives caused by linkage disequilibrium (LD), we further applied the HEIDI test, setting the threshold at *p* > 0.01. Appendix A shows the results of SMR analysis, which shows that several genes are causally associated with prostate cancer. Figure 2A shows the gene–trait association loci map within the same locus; *CASP8*, CASP10, and CFLAR were mapped to the ‘chr2:200659120:201800114’ locus and were associated with prostate cancer, which may indicate that the genetic association of prostate cancer with *CASP8* may be influenced by CASP10 and CFLAR. In the tumor microenvironment, the co-expression patterns of *CASP8*, CASP10, and CFLAR may influence tumor cell resistance to apoptosis and immune evasion. Understanding the co-localization and functional relationships among *CASP8*, CASP10, and CFLAR could provide new targets for immunotherapy. And CFLAR, a c-FLIP protein family member, as a truncated form of *CASP8* inhibits cell death, promotes tumorigenesis, and is associated with poor prognosis, as demonstrated [26]. Studies have reported that *CASP8* and CASP10 are key executioners in the apoptosis pathway, and their functional loss or reduced expression may lead to tumor cell resistance to apoptosis-inducing therapies, such as chemotherapy or radiotherapy [27]. CFLAR, as a negative regulator of *CASP8*, may inhibit the apoptosis pathway through its high expression, thereby enhancing tumor cell resistance to treatment [28].

### 3.3. Single-Cell Expression Patterns of CASP8 in Prostate Normal and Tumor Tissues

We analyzed the preprocessed data in Section 2.5, and after filtering out the abnormal cells, the remaining cells were divided into 25 cell clusters and further assigned into seven cell types according to the marker genes. Upon analysis, *CASP8* expression was found to be significantly higher in tumor tissues than in normal tissues, especially in T cells (Figure 2B,C). Subsequently, we subdivided the T cells into *CASP8* high-expressing T cells and *CASP8* low-expressing T cells according to the median value, and we found that the proportion of *CASP8* high-expressing T cells was much higher than that of normal cells in tumor tissues but still lower than that of *CASP8* low-expressing T cells (Figure 2D). Enrichment analysis indicated that *CASP8* high-expressing T cells had an upregulated oxidative phosphorylation pathway (Figure 2E).

### 3.4. Spatial Localization and Functional Implications of CASP8 in Prostate Cancer

This analysis is essentially the same as the gene set analysis, RCTD deconvolution information can be found in Appendix A. First, we found that *CASP8* expression at the single-cell level was concentrated in T cells (Figure 3A) and divided them into *CASP8*-positive T cells (*CASP8*+T) and *CASP8*-negative T cells (*CASP8*-T). Subsequently, they were analyzed for cellular communication. As shown in Figure 3B,C, there was a strong cellular communication between *CASP8*+ T cells and monocytes, with a predominance of HLA-CD8 interactions. Reverse chronological analysis revealed that *CASP8*+ T cells were located at the developmental end point of prostate cancer cells. The developmental end point of cells signifies that they have completed specific differentiation or functional programs and entered a terminal state, which is often associated with cellular senescence or functional exhaustion. This observation suggests that *CASP8* expression was reduced in dedifferentiated tumor cells, indicating a possible negative correlation between *CASP8* and cancer progression (Figure 3D,E).

As shown in Figure 3F, the spatial distribution characteristics of *CASP8*+ T cells and monocytes were similar. The correlation heatmap further indicated that *CASP8*+ T cells had a strong correlation with monocytes (Figure 3G). Misty co-localization analysis showed that *CASP8*- T and *CASP8*+ T had the highest R2 benefit (Figure 3H); the distributions of various cell types on spatial scales were dominated by intracellular ratios (Figure 3I); the spatial dependence of *CASP8*+ T cells on monocytes was significantly increased within juxta_5 and with increasing distance, the spatial dependence between *CASP8*+ T cells and monocytes increased and vice versa for *CASP8*- T (Figure 3J). Homotypic and heterotypic cell networks were seen as regions of interactions between *CASP8*+ T cells and between *CASP8*+ T and monocytes (Figure 3K).

### 3.5. Construction and Validation of BCR Prognostic Model for Prostate Cancer

We downloaded to obtain the gene expression matrix of the GSE116918 dataset and constructed the BCR prognostic model of prostate cancer based on the differential genes between *CASP8*+T and *CASP8*-T cells. The dataset was first randomly divided into training and test sets according to a 1:1 ratio, and then the training set was trained. One-way Cox regression screened nine genes associated with prognosis (*p* < 0.05), and then lasso–Cox regression was applied to combine lasso regression with Cox proportional risk regression to screen key variables predicting survival outcomes, and multifactorial Cox regression was used to further confirm whether the associations of the variables with survival were independent (Figure 4A–C). Eventually, four genes were screened as the core genes for constructing the BCR prognostic model for prostate cancer. The risk score formula was as follows: riskscore = 0.82 × TXNIP + 1.57 × XCL1 − 1.09 × KLF6 − 1.76 × SAMSN1. The prognostic model was able to differentiate between high-risk and low-risk groups well for the training set, the test set, and the entire dataset (Figure 4D). Time-dependent ROC analysis proved the model’s discriminatory ability throughout the entire dataset: the 5-year AUC for the full sample was 0.713. To further improve the predictive performance of the model, we combined the risk score and clinical parameters to construct a nomogram to predict the incidence of BCR at 3, 4, and 5 years after therapy for prostate cancer, and the results showed that the C-index was 0.683, with a certain degree of accuracy, and risk scores were independently associated with BCR (Figure 4E); and the calibration curves suggested a good predictive accuracy (Figure 4F). Decision curve analysis (DCA) was performed to calculate the clinical indicators, risk scores, and the net clinical benefit of the nomogram (Figure 4G). Immunohistochemical results suggested that in tumor tissues, *CASP8* and SAMSN1 showed high staining, TXNIP showed medium staining, while KLF6 and XCL1 showed negative staining; in normal tissues, KLF6 showed medium staining, *CASP8* and SAMSN1 showed low staining, while TXNIP showed negative staining (Figure 4H).

### 3.6. Clinical Subgroup Analysis

In order to further validate the clinical applicability of the prognostic model, we analyzed the difference in survival prognosis for different subtypes of clinical parameters. The results showed that the prognostic model was able to significantly differentiate between high- and low-risk groups under the conditions of age ≤ 70 or >70 years, Gleason >7, PSA >10, and T3–4, i.e., it had a more accurate differentiation for high-risk prostate cancer (Figure 5A–D).

### 3.7. Tumor Microenvironment Regulation, Immune Checkpoint Analysis, and Immunotherapy

Tumor stromal cells and immune cells also play an important role in tumor prognosis. In order to comprehensively assess the tumor microenvironment, we applied the “ESTIMATE” package to calculate the stromal score, immune score, and ESTIMATE score and inferred the tumor purity between the BCR high-risk and low-risk groups of prostate cancer. The results suggested that the high-risk group had lower stromal and immune cell components and showed higher tumor purity (Figure 6A). GSEA results suggested that the activated oxidative phosphorylation pathway was enriched in the high-risk group (Figure 6B). The tumor–immunity cycle is the process by which the body’s immune system responds to tumor antigens and kills tumor cells [29]. Specifically in prostate cancer, biallelic inactivation of CDK12 is associated with a unique genome instability phenotype. The CDK12-specific focal tandem duplications can lead to the differential expression of oncogenic drivers, such as CCND1 and CDK4. As such, there is a possibility of vulnerability to CDK4/6 inhibitors for CDK12-mutated tumors. Moreover, the CDK12 aberrations may be used next to mismatch repair deficiency as a biomarker of treatment response. This highlights the rationale for the combination therapeutic strategy of immune checkpoint blockade and CDK4/6 inhibition in clinical trials [30]. In cancer patients, this cyclic process may be impaired, thus leading to the body’s inability to generate antitumor immunity, which further contributes to the development of cancer. As shown in Figure 6C, high-risk patients have lower scores in priming and activation processes compared to the low-risk group, which further explains the poor prognosis of high-risk patients. Checkpoint analysis showed that SIRPA was downregulated in the high-risk group compared to the low-risk group, which may be another reason for the poor prognosis of this group (Figure 6D). Based on the GDSC database, we analyzed the sensitivity analysis of the different drug treatments between the high- and low-risk groups. The results, as shown in Figure 6E, showed that the sensitivity to olaparib was higher in the high-risk group compared to the low-risk group, which could help personalized treatment of prostate cancer.

## 4. Discussion

The study of Anoikis has unveiled another major mechanism of cancer metastasis, which provides a theoretical basis for our study of cancer progression and further provides value for cancer classification and treatment. We used multi-omics studies to analyze the spatial-specific expression of Anoikis genes in prostate cancer and its role in prognosis and treatment.

Caspases are classified into inflammatory caspases (caspase-1, -4, -5, and -11) and apoptotic caspases (caspase-2, -8, -9, and -10) [31]. *CASP8*, or caspase-8, plays an important role in inflammatory signaling and apoptosis [32]. Lorenzo Galluzzi et al. noted that *CASP8* regulates pro-inflammatory and anti-inflammatory processes in a variety of ways [31]. Ranadip Mandal et al. summarized the ability of *CASP8* to act as a “double-edged sword” in cancer and suggested that the elevated expression of *CASP8* in cancer may be due to its nonapoptotic progrowth function [33]. *CASP8* expression was significantly higher in prostate cancer tissues than in normal tissues and was concentrated in T cells, which may be due to its nonapoptotic function. *CASP8*, as a member of the cysteoaspartic enzyme family, has dual roles in cell death and survival; its nonapoptotic function confers resistance to enzalutamide in prostate cancer; and its high expression is often associated with poor prognosis of prostate cancer [34], which is consistent with our SMR results. It has been documented that *CASP8* expression is elevated in breast and pancreatic cancer cells [35] and correlates with poor prognosis in patients with hepatocellular carcinoma [36]. The nuclear localization of *CASP8* supports the potential of its nonapoptotic function [36,37,38]. As for prostate cancer, *CASP8* has been shown to have potential as a marker for high-risk prostate cancer [22]. TXNIP is involved in cellular oxidation, regulation of glucose metabolism and lipid metabolism, and tumor regulation, acting as an antitumor or protumor agent [39]. Qing Zhou et al. performed knockdown experiments and found that DPEP inhibited the process of glucose metabolism by inducing TXNIP, which led to the death of cancer cells [40]. Various studies have shown that TXNIP can play an anticancer role as a tumor suppressor in prostate cancer [41,42]. XCL1 has been less studied in prostate cancer, but existing studies have demonstrated that XCL1 is involved in the antitumor process and correlates with a good prognosis of the cancer [43,44,45]. KLF6 is a member of the Sp1/KLF family of transcription factors, which plays a role in cell development, differentiation, proliferation, and apoptosis [46]. In prostate cancer, KLF6 is considered a tissue-specific structure-specific prognostic marker for prostate cancer [47]; inactivation or overexpression of KLF6 as a tumor suppressor produces a protumorigenic effect and facilitates tumor progression [46,48,49]. SAMSN1 (HACS1), a member of the family of intracellular adaptor proteins encoding the SLY family, is often expressed in hematopoietic tissues [50]. This gene has been shown to be involved in B-cell activation and differentiation [51] and has been suggested as a possible tumor suppressor gene in diseases such as lung cancer and myeloma [52,53], but little is known about its role in prostate cancer.

There is growing evidence that metabolic reprogramming is important in cancer progression [54,55]. High glycolytic processes are considered to be a major feature of cancer cells. It is now widely recognized that cancer cells mainly rely on aerobic glycolysis to produce ATP for energy requirements, leading to an attenuation of the oxidative phosphorylation process, which is often referred to as the “Warburg effect” [56]. Recently, bioinformatics analysis has shown that a subpopulation of circulating tumor cells with high oxidative phosphorylation is associated with cancer progression in many types of cancers (including melanoma, prostate cancer, etc.), and this phenomenon has been described as the “Anti-Warburg Effect” and has been experimentally demonstrated in melanoma [57,58]. Our results showed that the high-risk tumor group had a stronger oxidative phosphorylation process, which further proved the reliability of our GSEA analysis. The discovery of the “Anti-Warburg Effect” also provides a new perspective on the treatment of prostate cancer progression.

BCR of prostate cancer is an event of particular concern after treatment of this tumor; BCR usually precedes clinical recurrence and is a predictor of elevated risk of distant metastasis, prostate cancer-specific mortality, and overall mortality [59]. Therefore, the BCR stage of a patient’s disease is a critical period of disease treatment, and close follow-up and aggressive treatment of the patient can help to extend the benefits. A study on BCR after radical prostatectomy showed that Gleason score (≤7 or >8), time from surgery to biochemical recurrence (≤3 years or >3 years), and prostate cancer-specific antigen doubling time (PSADT) were the only significant independent risk factors for the time from BCR to prostate cancer-specific death [60]. Clinical correlation analyses of our prognostic model showed that BCR for prostate cancer targeting Gleason > 7 was able to better differentiate between high- and low-risk groups. The better performance of our model was also reflected by its more accurate prediction of BCR recurrence at 3, 4, and 5 years after treatment.

Our study found that SIRPA was downregulated in the high-risk group. CD7/SIRα is an important innate immune checkpoint in cancer, and one study found that high SIRPA expression was associated with a favorable response to anti-PD-1 therapy [61], which explains the downregulation of SIRPA in the high-risk group. Drug sensitivity analysis revealed that patients with high-risk prostate cancer BCR may be more sensitive to the PARP inhibitor olaparib, which has been demonstrated in clinical trials, and that the drug is particularly effective in BRCA-positive metastatic prostate cancer [62,63]. It is reported that patients with BRCA2 pathogenic sequence variants have increased levels of serum PSA at diagnosis, an increased proportion of high Gleason tumors, elevated rates of nodal and distant metastases, and high recurrence rates [64]. This suggests that stratification of prostate cancer patients based on signature genes characterizing *CASP8* T cells could help in selecting patients who would benefit from olaparib treatment, which is a subdivision of personalized treatment for prostate cancer.

Although we explored the expression characteristics and prognostic role of the *CASP8* gene in prostate cancer and achieved certain results, there are still limitations. First, relying on public data mining, we failed to further validate the reliability of the results by basic experiments; second, our study was based on a small sample size, which could not guarantee the wide applicability of the results. In conclusion, further experiments to explore the characteristics of the prostate cancer tumor microenvironment are necessary.

## 5. Conclusions

This study provides a comprehensive characterization of Anoikis genes and *CASP8* in prostate cancer. We developed and validated a prognostic marker containing four genes that can effectively stratify prostate cancer patients. Patient risk scores were correlated with various clinical indicators, tumor microenvironment cell scores, tumor–immune circulation, and immunotherapy efficacy. In conclusion, *CASP8* can be used as a potential biomarker for prostate cancer and provide an effective prognosticator for the risk grade of BCR in prostate cancer, facilitating early-stage personalized treatment and improving prognosis. However, large-scale cohort studies and experimental validation are still needed.

## Figures and Tables

**Figure 1 biomedicines-13-00661-f001:**
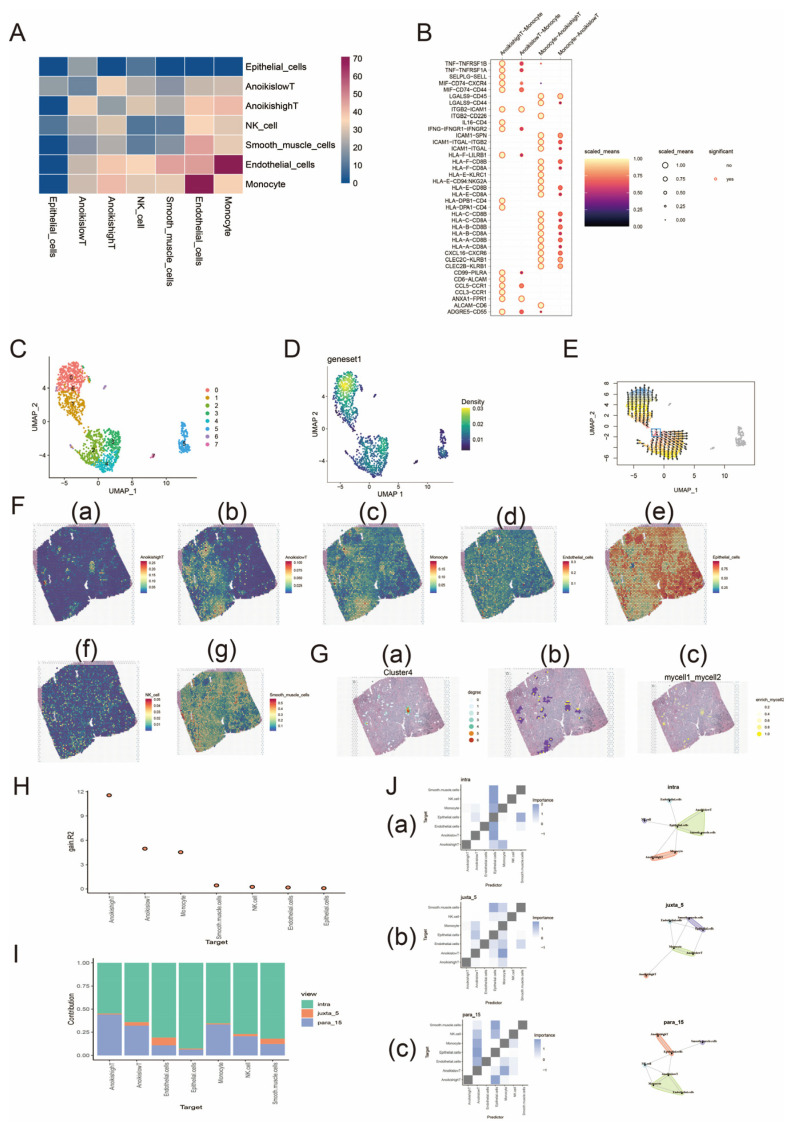
Single-cell and spatial transcriptome analyses of the Anoikis gene set in prostate cancer. (**A**) demonstrates the strength of cellular communication between the seven cell types. Colors from blue to red indicate a gradual increase in intensity. (**B**) demonstrates ligand–receptor interactions between AnoikishighT and monocytes. Darker to lighter colors and smaller to larger circles indicate a gradual increase in the strength of the interaction; circles with a border indicate a significant interaction and vice versa. (**C**) shows the UMAP downscaled images of the eight cell clusters. (**D**) shows the downscaled density map of the Anoikis gene set in T cells. The color from blue to yellow indicates a gradual increase in density. (**E**) illustrates the reverse chronological analysis of the Anoikis gene set in T cells. The position of the box represents the developmental starting point, and the arrow points to the developmental end point. In (**F**), (a)–(g) are the spatial expression characteristics of AnoikishighT, AnoikislowT, monocytes, endothelial cells, epithelial cells, NK cells, and smooth muscle cells, respectively. The homotypic cell network of AnoikishighT is shown in (**G**) (a). The degree from 0 to 6 indicates that the central cell is surrounded by a gradually increasing number of cells of the same type. (**G**) (b) shows the interactions region of the heterotypic cell network. (**G**) (c) is an enrichment scoring image of the interaction between AnoikishighT and monocytes. Darker colors represent stronger interactions. (**H**) demonstrates the degree of influence of spatial environment on different cell types (R2 benefit). (**I**) represents the proportion of predicted contribution of different spatial scales to cell types. Green indicates a spatial scale with radius 0; yellow indicates a spatial scale with radius 5; and blue indicates a spatial scale with radius 15. (**J**) demonstrates the co-expression analysis between cell subpopulations at different spatial scales. (a)–(c) show the analysis of different cell subpopulations at spatial scales of 0, 5, and 15, respectively. The figures shown in the left column are interaction heatmaps of cells at different spatial scales. The color gradient represents the strength of these interactions, with darker colors indicating stronger inter-actions. The images in the right column are community plots at different spatial scales. Each community represents a group of cells that are tightly distributed in space, and these cells may share functional similarities or collaborate in biological processes.

**Figure 2 biomedicines-13-00661-f002:**
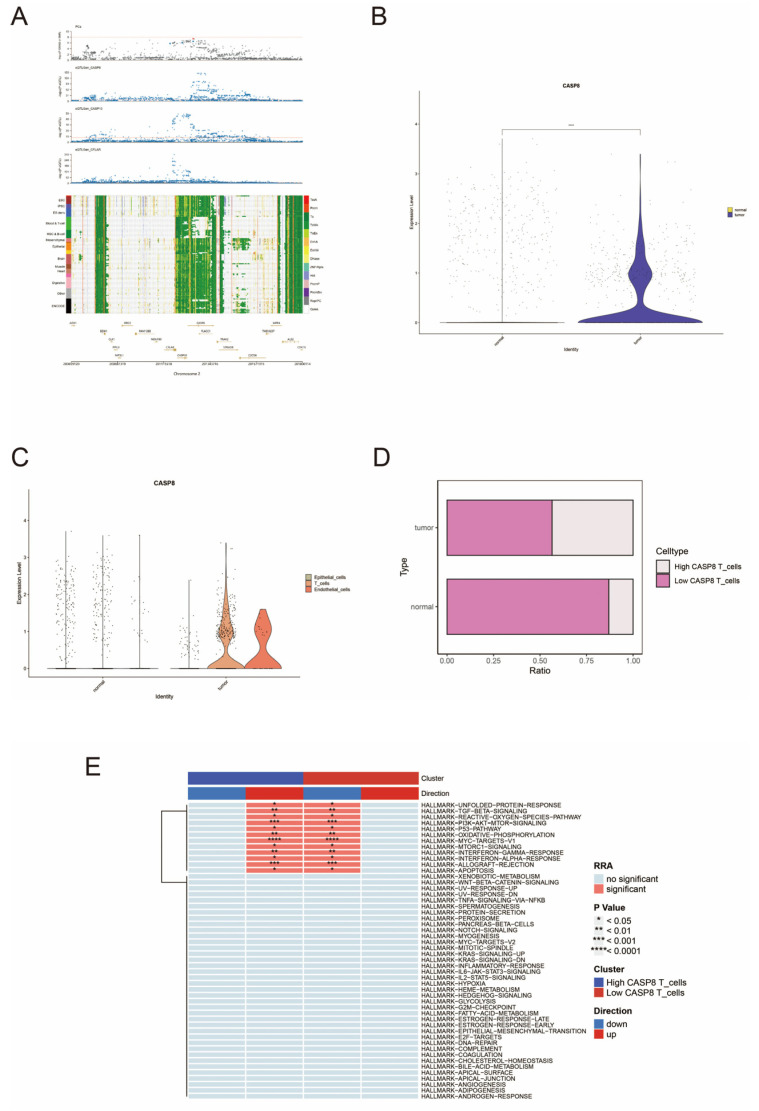
SMR analysis and single-cell analysis of *CASP8* in the prostate. (**A**) is a gene–trait association loci map from SMR analysis: images of association loci with prostate cancer for all genes within the same locus as *CASP8*. (**B**) shows the difference in expression levels of *CASP8* in tumor and normal tissues. Blue indicates tumor tissue and yellow indicates normal tissue. **** *p* < 0.0001. (**C**) demonstrates the expression difference of *CASP8* in different cells of tumor and normal tissues. Green represents epithelial cells; yellow represents T cells; orange represents endothelial cells. (**D**) shows the ratio of high and low *CASP8* expression in T cells within tumors and normal tissues. White indicates high *CASP8* expression, and pink indicates low *CASP8* expression. (**E**) demonstrates the HALLMARK pathway enrichment differences between high- and low-*CASP8*-expression groups in T cells. Blue in Cluster indicates the high expression of *CASP8*; red indicates the low expression of *CASP8*. Blue in Direction indicates downregulation and red indicates upregulation. * *p* < 0.05; ** *p* < 0.01; *** *p* < 0.001; **** *p* < 0.0001. No color in RRA indicates nonsignificant results; red indicates significant results.

**Figure 3 biomedicines-13-00661-f003:**
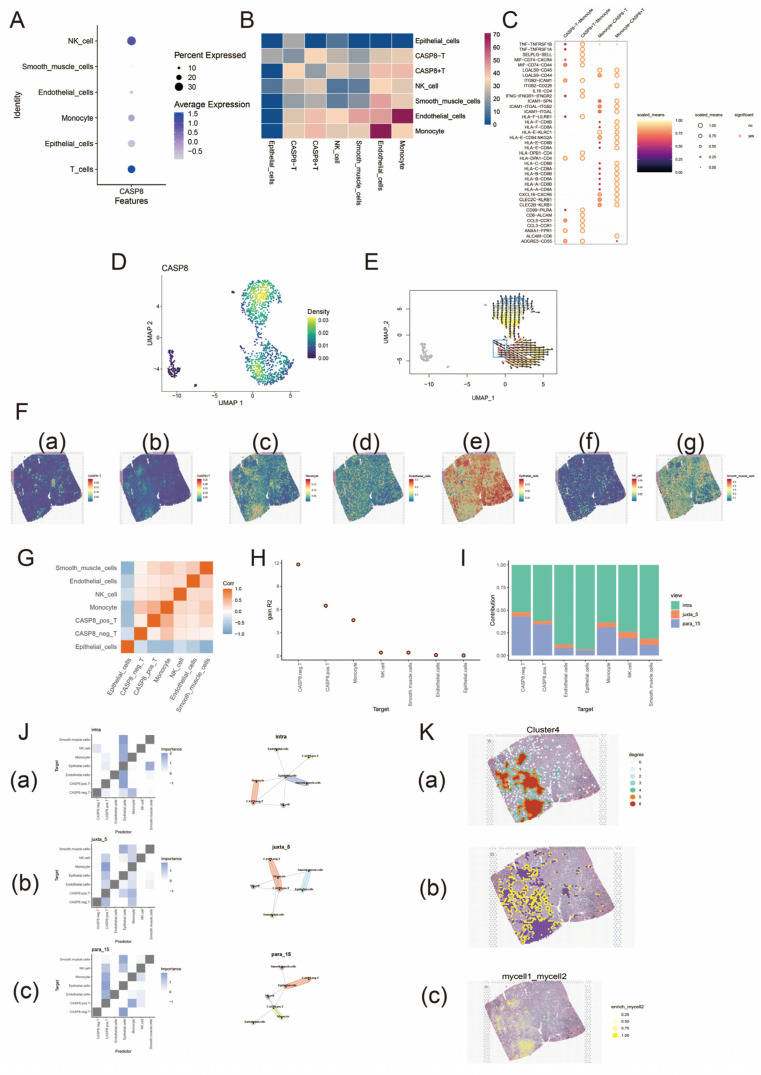
Single-cell and spatial transcriptome analysis of *CASP8* in prostate cancer. (**A**) shows the expression levels of *CASP8* in different cell types. The size of the circle represents the size of the expression ratio; the darker the blue color, the higher the expression level. (**B**) shows the intensity of cellular communication between the seven cell types. The blue to red color indicates a gradual increase in intensity. (**C**) shows the ligand–receptor interaction between *CASP8*+T and monocytes. Darker to lighter colors and smaller to larger circles indicate a gradual increase in the strength of the interaction; a bordered circle indicates a significant interaction and vice versa. (**D**) shows a downscaled density plot of *CASP8* in T cells. The blue to yellow color indicates a gradual increase in density. (**E**) shows the reverse chronological analysis of *CASP8* in T cells. The position of the box represents the developmental starting point and the arrow points to the developmental end point. The spatial expression characteristics of *CASP8*-T, *CASP8*+T, monocytes, endothelial cells, epithelial cells, NK cells, and smooth muscle cells are shown in (**F**) (a)–(g), respectively. The transition from blue to red indicates a gradual increase in the expression of the cell type in that spatial region. (**G**) shows the correlation analysis between different cell types. Blue color indicates low correlation and orange color indicates high correlation. (**H**) shows the degree of influence of spatial environment on different cell types (R2 benefit). (**I**) indicates the proportion of predicted contribution of different spatial scales to cell types. Green indicates a spatial scale of radius 0; yellow indicates a spatial scale of radius 5; and blue indicates a spatial scale of radius 15. (**J**) shows the co-expression analysis between cell subpopulations at different spatial scales. (a)–(c) show the analysis of different cell subpopulations at spatial scales of 0, 5, and 15, respectively. The figures shown in the left column are interaction heatmaps of cells at different spatial scales. The color gradient represents the strength of these interactions, with darker colors indicating stronger inter-actions. The images in the right column are community plots at different spatial scales. Each community represents a group of cells that are tightly distributed in space, and these cells may share functional similarities or collaborate in biological processes. (**K**) (a) shows the homotypic cell network of *CASP8*+T cells. From 0 to 6 indicates that the central cell is surrounded by progressively more homotypic cells. (**K**) (b) shows the interaction region of the heterotypic cell network between *CASP8*+T cells and monocytes. (**K**) (c) is an image of the enrichment score of *CASP8*+T cells interacting with monocytes. Darker colors represent stronger interactions.

**Figure 4 biomedicines-13-00661-f004:**
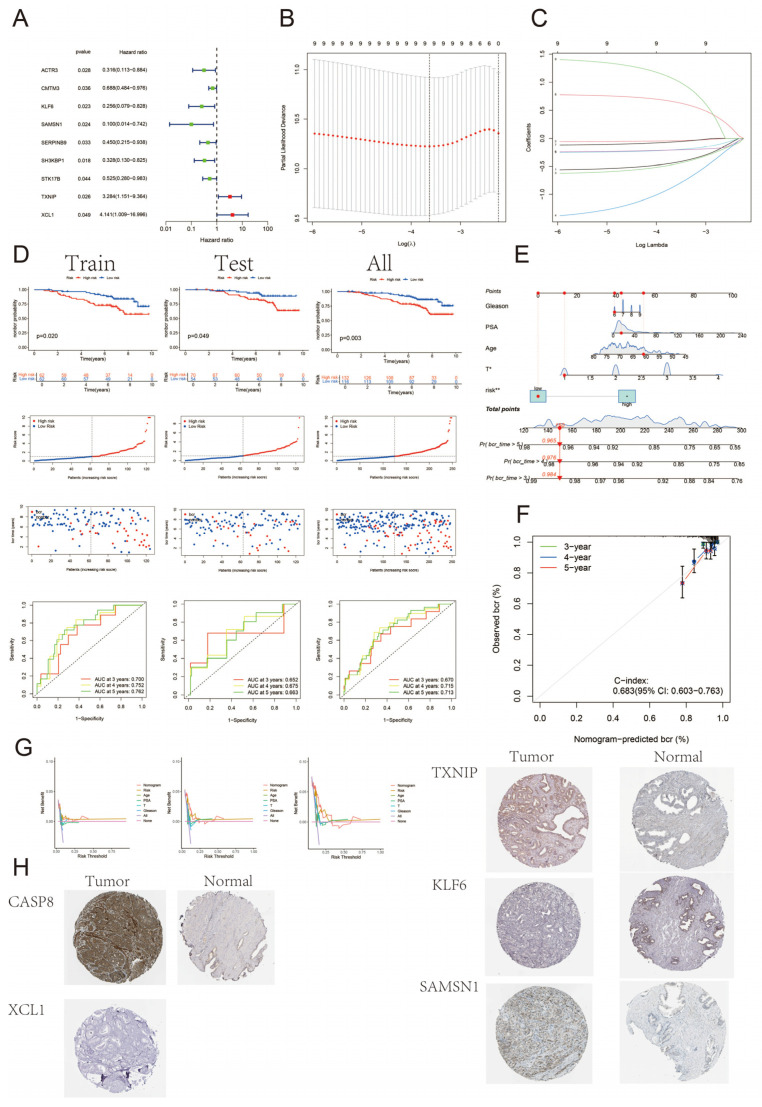
Construction of a BCR prognostic model for prostate cancer. (**A**) shows a schematic of the results of the one-way Cox regression analysis. (**B**,**C**) are lasso regressions for feature selection. (**D**) shows the survival analysis of the training set, test set, and full sample; risk score curve and scatter plot; and time-based ROC curve. (**E**) is the nomogram of 3-, 4-, and 5-year BCR probabilities used for prediction. (**F**) is the calibration curve for the nomogram. The diagonal line represents the perfect prediction of the ideal model. The colored solid line represents the agreement between the nomogram-predicted BCR probability and the actual probability for a given follow-up period. (**G**) shows the decision curve analysis of nomogram, risk score, and each clinical indicator. (**H**) shows the immunohistochemical results of *CASP8* and core genes in tumor and normal tissues (scale bars: 200 μm).

**Figure 5 biomedicines-13-00661-f005:**
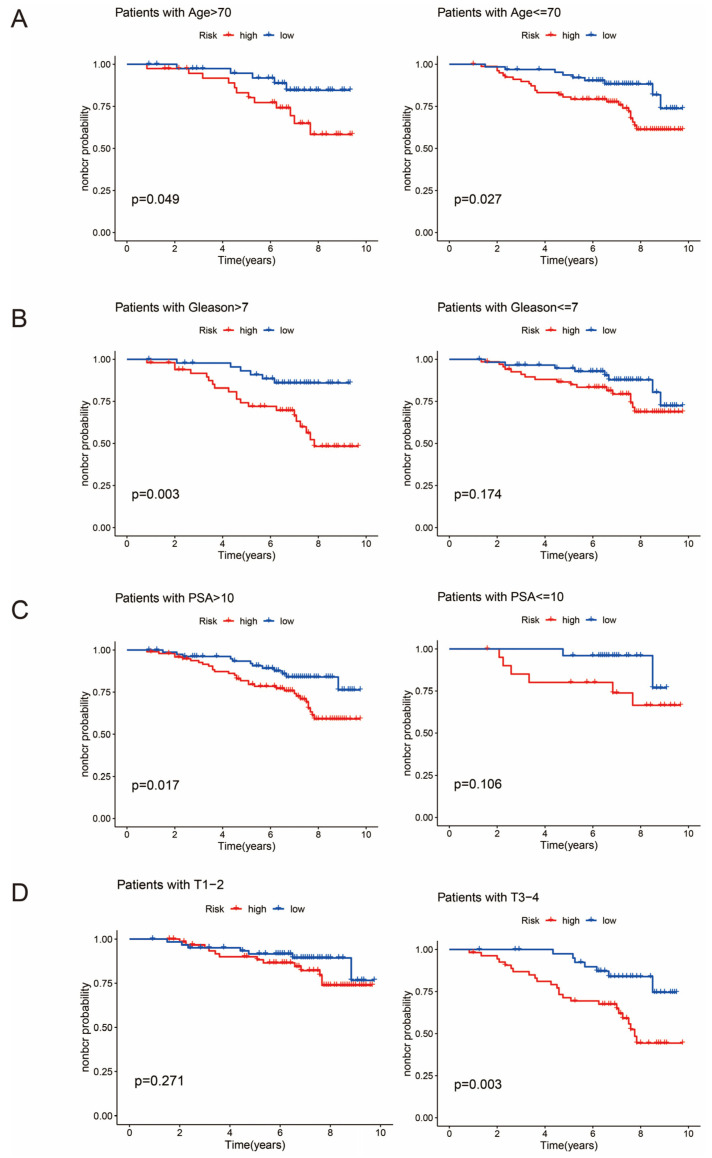
Clinical subgroup analysis of prognostic models. (**A**) shows a comparison of survival analyses between age ≤ 70 and age > 70. (**B**) shows a comparison of survival analysis between Gleason ≤ 7 and Gleason > 7. (**C**) shows the comparison of survival analysis between PSA ≤ 10 and PSA > 10. (**D**) shows the comparison of survival analysis between stages T1–2 and T3–4.

**Figure 6 biomedicines-13-00661-f006:**
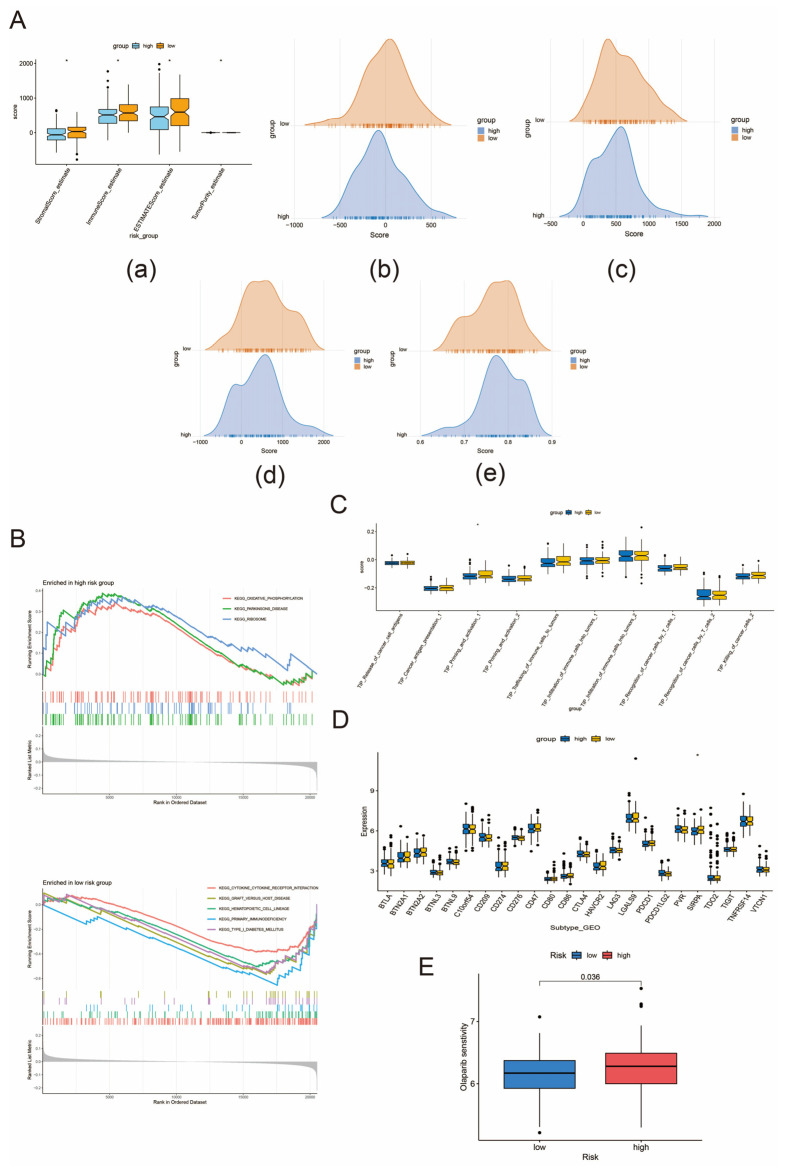
Tumor microenvironment, GSEA, tumor–immune circulation, immune checkpoints, and drug sensitivity analysis between high- and low-risk groups. (**A**) shows the tumor microenvironment analysis between high- and low-risk groups. (a) is the analysis of the difference in stromal score, immune score, ESTIMATE score, and tumor purity between the high- and low-risk groups. (b)–(e) are peak plots of stromal score, immune score, ESTIMATE score, and tumor purity between high- and low-risk groups, respectively. (**B**) shows the GSEA analysis of the high- and low-risk groups. A convex upward curve indicates upregulation of the pathway and a convex downward curve indicates downregulation of the pathway. (**C**) is the tumor–immune circulation analysis of the high- and low-risk groups. * indicates that key steps in the cycle differ in the high- and low-risk groups. Blue color indicates high-risk group and yellow color indicates low risk. (**D**) is an immune checkpoint analysis of the high- and low-risk groups. * indicates that checkpoints are significantly different between high- and low-risk groups. Blue color indicates high-risk group and yellow color indicates low-risk group. (**E**) shows the drug sensitivity analysis between the high- and low-risk groups. Blue color indicates low-risk group and red color indicates high-risk group.

## Data Availability

No raw data were generated for this study. The sources of data used are cited in the text. All other relevant data are included in the main text.

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
