# Peer review of "Multi-Omics Analysis of the Anoikis Gene CASP8 in Prostate Cancer and Biochemical Recurrence (BCR)"

_biomedicines, 2025, doi:10.3390/biomedicines13030661_

Round 1
Reviewer 1 Report
Comments and Suggestions for Authors
In this study, the authors present a prognostic model for the high and low risk groups of prostate cancer BCR by characterizing the signature genes of CASP8+ T cells and explored the clinical applicability of the model, which can help to guide clinical decision-making and provide personalized treatment for patients. The manuscript is straightforward, well written, and concise and has clear results. Definitely deserves to be published and is a valuable contribution to the “Biomedicines” journal.
However, the following comments need to be addressed, as recommended.
[1] “1. Introduction:”, Page 1 of 15, Lines 32-33:
“As the second most common cancer in men nowadays, its early asymptomatic presentation and inert process are particularly prominent [2,3].”.
It is also worthy to discuss that there is significant interest in integrating image-based information from radiomics into the multiomics framework, aiming to combine biomolecular-level information with imaging data. The clinical application of radiomics involves identifying the relationship between the features extracted from images and the clinical outcome of interest. In prostate cancer patients, common imaging modalities include MRI, transrectal ultrasound, conventional CT, cone-beam CT, and molecular imaging, often in the form of PET/CT with tracers such as radiolabeled Prostate-Specific Membrane Antigen (PSMA) and fluorine-labeled 18F-choline.
Recommended reference: Tapper W, et al. The Application of Radiomics and AI to Molecular Imaging for Prostate Cancer. J Pers Med. 2024;14(3):287.
[2] “1. Introduction:”, Page 1 of 15, Lines 33-35:
“With the recent development of modern technology, the treatment of prostate cancer has developed a systematic and individualized process.”.
At that stage, the authors should report that abiraterone with prednisolone combined with androgen deprivation therapy (ADT) should be considered a new standard treatment for patients with high-risk non-metastatic prostate cancer. In metastatic setting, enzalutamide and abiraterone should not be combined for those starting long-term ADT. Clinically important improvements in survival from addition of abiraterone to ADT are maintained for longer than 7 years.
Recommended reference: Attard G, et al. Abiraterone acetate plus prednisolone with or without enzalutamide for patients with metastatic prostate cancer starting androgen deprivation therapy: final results from two randomised phase 3 trials of the STAMPEDE platform protocol. Lancet Oncol. 2023;24(5):443-456.
[3] “3.7. Tumor microenvironment regulation, immune checkpoint analysis and immunotherapy”, Page 9 of 15, Lines 389-390:
“Tumor-immunity cycle is the process by which the body's immune system responds to tumor antigens and kills tumor cells [25].”.
Specifically in prostate cancer, biallelic inactivation of CDK12 is associated with a unique genome instability phenotype. The CDK12-specific focal tandem duplications can lead to the differential expression of oncogenic drivers, such as CCND1 and CDK4. As such, there is a possibility of vulnerability to CDK4/6 inhibitors for CDK12-mutated tumors. Moreover, the CDK12 aberrations may be used next to mismatch repair deficiency, as a biomarker of treatment response. The authors should explain that this highlights the rationale for the combination therapeutic strategy of immune checkpoint blockade and CDK4/6 inhibition in clinical trials.
Recommended reference: Boussios S, et al. Aberrations of DNA repair pathways in prostate cancer: a cornerstone of precision oncology. Expert Opin Ther Targets. 2021;25:329-333.
[4] “4. Disscusion”, Page 11 of 15, Lines 485-488:
“Drug sensitivity analysis revealed that patients with high-risk prostate cancer BCR may be more sensitive to the PARP inhibitor olaparib, which has been demonstrated in clinical trials, and that the drug is particularly effective in BRCA-positive metastatic prostate cancer [57,58].”.
At that point, the authors should also report that patients with BRCA2 pathogenic sequence variants have increased levels of serum PSA at diagnosis, an increased proportion of high Gleason tumors, elevated rates of nodal and distant metastases, and high recurrence rates.
Recommended reference: Shah S, et al. BRCA Mutations in Prostate Cancer: Assessment, Implications and Treatment Considerations. Int J Mol Sci. 2021;22(23):12628.
Minor comment:
Please revise the reference list to align with the journal's styling requirements.
Author Response
As per your request, I have added the relevant content in the corresponding sections, highlighted them in red, and included citations to the related literature. Additionally, we have also revised the formatting of the reference list.
Reviewer 2 Report
Comments and Suggestions for Authors
Abstract
1. Clearly highlight the novelty of exploring CASP8-positive T cells in prostate cancer recurrence.
2. Quantify key results, such as predictive accuracy or hazard ratios from the prognostic model.
3. Replace "The occurrence of BCR is a challenge after early prostate cancer treatment" with "BCR remains a significant challenge following early prostate cancer treatment."
4. Simplify "The results revealed that high-risk prostate cancer BCR patients had various characteristics" to "High-risk prostate cancer BCR patients exhibited distinct characteristics."
5. Briefly mention the clinical implications of targeting CASP8 for personalized treatment strategies.
Introduction
1. Provide a more detailed rationale for selecting CASP8 as the focus gene for Anoikis-related research.
2. Include global statistics on prostate cancer recurrence rates to emphasize the study's significance.
3. Simplify "Prostate cancer is an androgen-dependent malignant tumor with high male prevalence" to "Prostate cancer is a common androgen-dependent malignancy in men."
4. Revise "The occurrence of BCR after treatment is still troubling" to "BCR following treatment remains a concern."
5. Compare CASP8's role in prostate cancer to other cancers where it has been implicated in cell death and tumor growth.
Methods
1. Provide justifications for selecting specific datasets (GSE193337 and GSE116918) and their relevance to prostate cancer.
2. Elaborate on the criteria for filtering cells during single-cell analysis.
3. Simplify "We grouped the T cells in the preprocessed data of tumor samples based on gene set scores into 2 groups" to "T cells were grouped into two categories based on gene set scores."
4. Clarify why the “LogNormalize” function and UMAP algorithm were selected for dimensionality reduction and visualization.
Results
Single-cell and Spatial Transcriptome Analysis
1. Explain the significance of the HLA-CD8 interaction between monocytes and Anoikis-high T cells.
2. Provide statistical significance for the observed spatial patterns.
3. Replace "We further analyzed the ligand-receptor interactions between T cells and monocytes" with "Ligand-receptor interactions between T cells and monocytes were further analyzed."
4. Discuss the implications of Anoikis-high T cells being located at the developmental endpoint.
SMR Analysis
1. Specify the p-values and thresholds used to identify causal relationships between eQTLs and prostate cancer.
2. Clarify the implications of CASP8’s co-location with CASP10 and CFLAR.
3. Replace "we found that CASP8, CASP10, and CFLAR were located in the 'chr2:200659120:201800114' locus" with "CASP8, CASP10, and CFLAR were mapped to the 'chr2:200659120:201800114' locus."
4. Include a discussion on whether the observed genetic interactions might influence therapeutic resistance.
Discussion
1. Critically evaluate the limitations of relying solely on public datasets without experimental validation.
2. Provide recommendations for validating the four-gene prognostic model in future studies.
3. Simplify "Our study showed that CASP8 was highly expressed in prostate cancer tissues compared to normal tissues" to "CASP8 expression was significantly higher in prostate cancer tissues than in normal tissues."
4. Discuss potential therapeutic interventions targeting CASP8 and related pathways.
Conclusion
1. Summarize the clinical relevance of CASP8 as a prognostic biomarker.
2. Suggest future applications of the prognostic model for guiding treatment decisions.
3. Replace "thus providing personalized treatment to patients at an early stage and improving prognosis" with "facilitating early-stage personalized treatment and improving prognosis."
4. Emphasize the need for larger cohort studies and experimental validation.
Comments on the Quality of English Language
The English could be improved to more clearly express the research.
Author Response
Abstract: The revised content has been highlighted in red within the relevant section.
Introduction: The revised content has been highlighted in red within the relevant section, excluding the highlighted portions from the latter half of line 4 to line 11 in the first paragraph, and from line 13 to line 18.
Methods: comment1:The GSE193337 dataset contains scRNA-seq data of prostate cancer, which can be used for single-cell analysis, while the GSE116918 dataset includes gene expression matrices of prostate cancer, suitable for subsequent genomic analysis.
Comments2-3:The revised content has been highlighted in red in Section 2.2.
Comment4:The "LogNormalize" function normalizes the gene expression values for each cell, typically by dividing the expression values of each cell by the total expression count (or another normalization factor), followed by a logarithmic transformation. This stabilizes the variance of the data and reduces the impact of extreme values.
UMAP is a nonlinear dimensionality reduction algorithm particularly well-suited for processing high-dimensional single-cell data. It preserves both local and global structures of the data while maintaining high computational efficiency, making it suitable for large-scale single-cell datasets.
Results:
Single-cell and Spatial Transcriptome Analysis
Comments1,3,4: The revised content has been highlighted in red in the Results section, excluding Sections 3.2 and 3.7.
Comment2: We thank the reviewer for their valuable comments. We understand the importance of statistical significance analysis for validating spatial patterns, but the primary goal of this study is exploratory analysis, focusing on revealing the correlations in single-cell spatial distribution and their potential biological significance.
SMR Analysis
The revised content has been highlighted in red in Section 3.2.
Discussion:
Comments1,2,4: We believe that this point has been addressed, either directly or indirectly, in the discussion section, and therefore no further elaboration is necessary. We thank the reviewer for their suggestion.
Comment3: The revised content has been highlighted in red.
Conclusion:
The revised content has been highlighted in red.
Round 2
Reviewer 1 Report
Comments and Suggestions for Authors
The authors have successfully addressed my comments.
the revised manuscript should be accepted for publication.
Reviewer 2 Report
Comments and Suggestions for Authors
Accept in present form